# A Frail Hairy Cell Leukemia Patient Successfully Treated with Pegylated Interferon-α-2A

**DOI:** 10.3390/jcm12010193

**Published:** 2022-12-27

**Authors:** Danilo De Novellis, Valentina Giudice, Vincenzo Ciccone, Paola Erra, Alba De Vita, Francesca Picone, Bianca Serio, Carmine Selleri

**Affiliations:** 1Department of Medicine, Surgery and Dentistry, “Scuola Medica Salernitana”, University of Salerno, 84081 Baronissi, Italy; 2Hematology and Transplant Center, University Hospital “San Giovanni di Dio e Ruggi D’Aragona”, 84131 Salerno, Italy; 3Radiology Unit, University Hospital “San Giovanni di Dio e Ruggi D’Aragona”, 84131 Salerno, Italy

**Keywords:** frailty, hairy cell leukemia, interferon

## Abstract

Hairy cell leukemia (HCL) treatment in elderly, frail subjects is still unsatisfactory, and interferons, old-fashioned therapies, can be effectively used in this subset of patients. Here, to the best of our knowledge, we report for the first time an old, frail HCL patient effectively and safely treated with pegylated interferon-α-2a in monotherapy as a first-line treatment. At diagnosis, the patient arrived in a life-threating condition due to severe neutropenia and splenomegaly with high risk of splenic rupture. However, splenectomy was proposed and refused by the patient; therefore, a therapy with pegylated interferon-α-2a was initiated. After six months of therapy, the patient displayed the disappearance of palpable splenomegaly and of peripheral hairy cells at morphological examination without any drug-related adverse event. Our case report supports the use of pegylated interferon-α-2a in monotherapy as an effective and safe alternative therapeutic option in frail, elderly patients not eligible for purine analogous or targeted therapies.

## 1. Introduction

Hairy cell leukemia (HCL), a rare, indolent, clonal B-cell disorder, is characterized by the presence of hairy cells with cytoplasmic projections accumulating in the bone marrow (BM), spleen, and peripheral blood, ultimately leading to BM fibrosis, peripheral cytopenia(s), and splenomegaly [1,2]. Hairy cells are larger than normal lymphocytes and usually have a round or a kidney-shaped nucleus, loose chromatin, pale cytoplasm, and peripheral protrusions. In most cases, neoplastic clones express CD19, CD20, CD22, CD11c, CD25, CD79b, CD103, CD123, and FMC7 and are usually negative for CD5, CD10, and CD23 [1,2,3,4]. Hairy cells also show monoclonal light-chain immunoglobulin (SmIg) restriction and frequently harbor a recurrent V600E somatic mutation on the B-raf proto-oncogene (*BRAF*), leading to Ras-independent constitutive activation of the RAF/MEK/ERK pathways and promotion of cell proliferation and survival [5,6].

Historical therapeutic strategies, including corticosteroids, alkylating agents, standard chemotherapy, splenectomy, or recombinant interferon-α, have been replaced by purine analogues, such as cladribine and pentostatin, which have become the first therapeutic choice in fit and young HCL patients [7,8,9,10,11]. Moreover, RAF inhibitor vemurafenib is a promising targeted therapy in relapsed/refractory HCL, especially in combination with rituximab, an anti-CD20 monoclonal antibody with a manageable toxicity profile [12,13]. Several other novel, targeted therapies are under investigation for treatment of relapsed/refractory HCL patients, such as recombinant anti-CD22 immunotoxin moxetumomab pasudox, dabrafenib plus trametinib, and Bruton’s tyrosine kinase (BTK) inhibitors. Ibrutinib, an oral BTK inhibitor, administered as single agent, has shown prolonged clinical benefits in relapsed HCL patients with an overall response rate (ORR) of 54%, an estimated 36-month progression-free survival (PFS) rate of 73%, and an overall survival OS of 85% [14,15,16].

Interferon-α has been widely used as first-line treatment in the past, with clinical efficacy and safety [17,18]. To date, its use is recommended as a first line HCL therapy in pregnant women, in patients with severe neutropenia or those not eligible for purine analogues, and in relapsed/refractory HCL [4]. However, no data are available on the efficacy and safety of pegylated (PEG) interferon-α-2A for treatment of newly diagnosed, frail HCL patients. Here, we report a case of an old, frail HCL subject with severe neutropenia and splenomegaly who was successfully treated with weekly administrations of PEG interferon-α-2A.

## 2. Case Presentation

An 82-year-old male came under our observation in January 2022 because of progressive pancytopenia and a history of bacterial pneumonia requiring hospitalization and third-generation cephalosporin administration in December 2021. In July 2021, the patient already showed mild thrombocytopenia (platelet count, 97,000/µL) and mild neutropenia (white blood cells (WBC), 5900 cells/µL; and absolute neutrophil count (ANC), 1200 cells/µL), with slightly reduced hemoglobin levels (12.7 g/dL). In January 2022, he displayed a worsening of complete blood counts (CBCs) with normocytic anemia (hemoglobin levels, 9.3 g/dL; mean corpuscular volume (MCV), 97 fL); mild thrombocytopenia (97,000 platelets/µL) and severe neutropenia (WBC, 3700 cells/µL; and ANC, 300 cells/µL). At physical examination, his spleen was palpable at the left iliac fossa, and he reported pain and a postprandial fullness sensation at the left hypochondrium. No B symptoms (fever, pruritus, drenching night sweats, or loss of ≥10% of body weight over 6 months) were referred. Patient’s frailty was evaluated using the frailty index (FI), defined as FI score = Number of health deficits present/Number of health deficits measured, showing a mild frailty (FI score, 0.332) [19,20].

Morphological analysis was performed on peripheral blood, showing the presence of 50% lymphoid cells with intermediate–high nucleus/cytoplasm ratio, loose chromatin, rare nucleoli, basophil cytoplasm, and peripheral hairy protrusions (Figure 1).

Flow cytometry immunophenotyping was also performed on peripheral blood, evidencing a B-cell clonal population positive for CD20, CD22, CD25, CD11c, CD103, and CD123 and negative for CD5, CD10, and CD38 (60% of nucleated cells analyzed). No other alterations were observed, as the remaining 27% of lymphocytes was represented by: 78% CD3^+^ T lymphocytes; 13% CD19^+^ B cells with no κ/λ restriction (κ/λ, 61%/39%); 24% CD3^−^CD56^+^ natural killer (NK) cells; and 15% CD3^+^CD56^+^ NKT cells. CD14^+^ monocytes were slightly reduced (0.3% of total nucleated cells analyzed). 

V600E *BRAF* mutation was detected by next-generation sequencing (NGS). BM biopsy confirmed the presence of hairy cells (60% of total examined cells), and HCL diagnosis was made. Total-body CT scan was also performed, and severe splenomegaly (longitudinal diameter (LD), 21 cm; and transversal diameter (TD), 12 cm) was documented. Moreover, large, hypodense, irregular areas within splenic parenchyma were observed, suggestive of initial splenic rupture, supported by the presence of a peri-splenic fluid. Several lymph node enlargements were observed: at hepatic hilum (maximum diameter, 2.6 cm), interaortocaval, and left paraaortic areas (maximum diameter, 1.3 cm) (Figure 2A and Figure 3A).

Urgent splenectomy was proposed because of high risk of splenic rupture. However, the patient refused surgery, and an alternative therapeutic strategy was quickly decided. Purine analogues were contraindicated, because of recent pneumonia in the setting of severe neutropenia, older age, and comorbidities, including chronic atrial fibrillation, mild heart failure, and type 2 diabetes. Other treatment options were discussed with the patient, who expressed the will to avoid intravenous therapies; therefore, monoclonal antibodies were excluded. Moreover, tyrosine kinase inhibitors (TKIs) were contraindicated because of cardiovascular comorbidities. Consequently, subcutaneous PEG interferon-α-2A was chosen as therapeutic option and was started at a dose of 90 µg once a week with acetaminophen premedication. Antibiotic and antifungal prophylaxis were given with levofloxacin and fluconazole. Growth factors were not administered to favor ANC recovery because of severe splenomegaly and high risk of splenic rupture. 

After three months of therapy, a slight hematological improvement was observed with an increase in hemoglobin levels to 10.4 gr/dL, platelet count to 121,000/µL, and ANC to 830 cells/µL. After six months, hemoglobin levels were normalized (13 gr/dL), and platelets raised to 125,000/µL and ANC to 2230 cells/µL. At physical examination, the spleen was not palpable. No drug-related reactions requiring dose reduction or discontinuation adjustment were documented. Furthermore, no infectious complications were observed. Peripheral blood morphological examination did not show the presence of hairy cells, and the minimal residual disease (MRD) evaluated by multi-parametric flow cytometry was 0.75%. No alterations were observed in the lymphocyte subsets except for a decrease in CD19^+^ B cells and an increase in CD14^+^ monocytes (CD3^+^ T lymphocytes, 75% with CD4^+^/CD8^+^ cells, 55%/19%; NKT cells, 13%; CD19^+^ B cells, 3%; and CD14^+^ monocytes 7%). Decreased B cells after treatment might mirror a reduction in certain B cell subpopulations, such as B regulatory cells with immunomodulatory functions, and might be related to physiological immunological reconstitution, as documented in other hematological malignancies, including Hodgkin lymphoma [21].

CT scan re-evaluation was also performed, showing a partial reduction of involved lymph nodes: 23% at hepatic hilum (2 cm vs. 2.6 cm) and 46% of interaortocaval area (0.7 cm vs. 1.3 cm). A significant splenic reduction was documented: 43% reduction from 21 cm of LD at baseline to 12 cm after six months of therapy. Parenchymal hypodensity areas were also reduced, especially those present at the middle third at diagnosis (Figure 2B and Figure 3B). Disease-related symptoms disappeared with a significant improvement in quality of life. Therefore, a complete response, defined as disappearance of palpable splenomegaly and of peripheral hairy cells at morphological examination in the absence of BM aspiration or biopsy reassessment [22], was achieved after six months of PEG interferon-α-2A therapy. At the time of writing, after one year of therapy, the patient is still alive in complete response, and he is continuing to receive PEG interferon-α-2A administration. 

## 3. Discussion

To date, HCL, a B-cell indolent disease, has been effectively and safely treated with purine analogues in young and fit patients, while the treatment of older and unfit subjects remains unsatisfactory [4]. In our case, PEG interferon-α-2A was chosen as a first-line therapy because of its favorable toxicity profile. Indeed, our patient had severe neutropenia, high risk of splenic rupture, and several comorbidities, including cardiovascular diseases that contraindicated the use of TKIs, and advanced age.

The mechanism of action of interferon-α in HCL is still under debate and is likely related to the ability to induce cell differentiation, growth arrest, and apoptosis [22]. Interferon-α binds to surface receptors (e.g., IFNAR1), leading to the activation of Janus kinases (JAK)/signal transducer and activator of transcription (STAT) pathways and to expression of antiproliferative and proapoptotic genes, including caspases and Fas/CD95 [23]. Interferon-α can also directly inhibit hematopoietic stem cell growth, mainly by blocking JAK/STAT signaling, or indirectly by reducing the production of stimulatory factors, such as IL-11 [24]. These effects can reduce leukemic cell growth and also cause myelosuppression as a drug-related side reaction [23,24]. Interferon-α can promote tumor cell killing and clearance by CD8^+^ T cells, macrophages, and NK cells, increase the expression of tumor-associated antigens, and enhance antigen presentation by dendritic cells [24,25]. The first clinical reports on interferon efficacy in HCL date back to 1984, when hematologic remissions were demonstrated in seven patients, and interferon efficacy was subsequently confirmed in larger cohorts treated with different interferon types (recombinant interferon-α-2a, recombinant interferon-α-2b, or partially purified human leukocyte interferon-α) [26,27]. More recent, retrospective trials confirmed the efficacy and safety of interferon-α as an induction and maintenance HCL therapy, with an ORR of 82–94% and lasting complete response (CR) rates of 13–24%, with a good toxicity profile [18,27,28,29,30]. Interferon-α also showed high efficacy as a first-line therapy in patients aged ≥65 years with comorbidities, or in those resistant to purine analogues, with a 5-year PFS of 68–96% [30]. HCL patients treated with interferon-α have durable responses, even in follow-up periods longer than seven years [28,29,30,31]. Interferon-α in monotherapy can be used as a priming therapeutic phase followed by combination with rituximab with good tolerability even in immunosuppressed HCL patients [27,32]. During the priming phase, interferon-α-2a can increase CD20 antigen expression on hairy cells and simultaneously stimulate CD8^+^ T cell and NK cytotoxic activity. Therefore, when rituximab is combined with interferon-α-2a during the second phase, tumor cell killing is enhanced likely through antibody-dependent cell mediated cytotoxicity, as demonstrated by the clinical efficacy of interferon-α-2a and rituximab in low-grade lymphomas [27,32,33,34]. Reported discontinuation rates, due to unmanageable toxicities, are very low (<20%), even though interferon-α-2a treatment is potentially associated with hematological and extra-hematological side effects [31]. 

Our patient arrived under our observation in a life-threating condition due to recent pneumonia in the setting of severe neutropenia and severe splenomegaly with a high risk of rupture. Splenectomy, although not curative, was proposed and refused by the patient. Purine analogues were contraindicated because of the high risk of worsening of clinical conditions and because of the recent pneumonia, severe neutropenia, older age, and comorbidities. Our patient also refused intravenous medications, and monoclonal antibodies, including rituximab, could not be used for induction. Moreover, rituximab was not chosen, because of long-term B-cell deficiency, leading to several infections, including severe coronavirus-19 disease (COVID-19), even though rituximab in monotherapy or in combination with other drugs induces a faster disease remission and hematological recovery with acceptable toxicity [35,36,37]. Other targeted therapies, such as vemurafenib, anti-CD22 moxetumumab pasudox, and ibrutinib, are not approved in our country as a first-line treatment of HCL [36]. However, TKIs, including ibrutinib, were not indicated because of cardiovascular comorbidities, especially atrial fibrillation [16]. Even though PEG interferon-α-2A monotherapy potentially induces a slower and suboptimal disease response, we chose the best possible therapeutic strategy for this patient by balancing clinical efficacy and safety and long-term complications. For these reasons, we did not opt for a PEG interferon-α-2A and rituximab combination, differently from a previously reported case [27]; thus, treatment with PEG interferon-α-2A in monotherapy was initiated at the lowest weekly dosage (90 µg vs. 135 µg), based on previously published studies showing efficacy and safety in other hematological malignancies, such as myeloproliferative neoplasms [38]. Dose increase was not performed because the patient showed hematological response and good tolerability at that PEG interferon-α-2A dosage.

The pegylated form was chosen because of it has a more favorable pharmacokinetic profile and lower administration frequency than standard interferons, despite the absence of previously published literature showing the efficacy and safety of pegylated formulation in HCL [39]. Hematological side effects were closely monitored with weekly complete blood counts. Other extra-hematological toxicities were also monitored with periodic blood tests for endocrine and liver functions and for autoimmunity [40]. Flu-like effects were significantly reduced using a premedication with acetaminophen before PEG interferon-α-2A administration. In our case, we showed efficacy of PEG interferon-α-2A in inducing a hematological improvement and spleen size reduction, ultimately leading to an improvement in the patient’s quality of life. Pegylated formulations might be the best therapeutic option in unfit, older patients, because of their favorable pharmacokinetic profile, characterized by higher half-life, tissue distribution, and by a more stable and durable serum concentration [41]. However, limited data regarding optimal treatment duration are available, and the therapeutic strategy might be modulated based on clinical response and tolerance, considering the association of an additional agent, such as rituximab, in the event of refractoriness or relapse, and a reduction in dosage or frequency of administration or drug discontinuation can be considered if a profound and stable remission is achieved. 

In conclusion, we showed that a PEG interferon-α-2A formulation can be effectively and safely used in treatment of unfit, older HCL patients. However, further validation in larger prospective trials is needed.

## Figures and Tables

**Figure 1 jcm-12-00193-f001:**
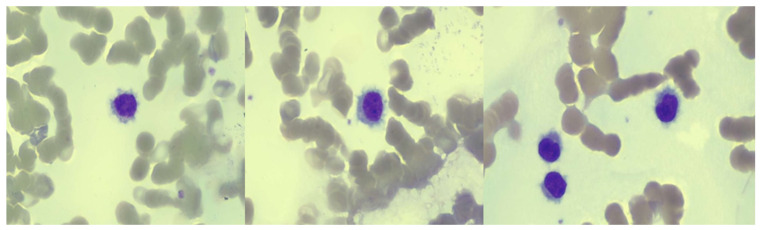
Morphological examination of peripheral blood smear shows the presence a hairy cell population with cytoplasm protrusions (May–Grünwald–Giemsa staining, original magnification, ×40).

**Figure 2 jcm-12-00193-f002:**
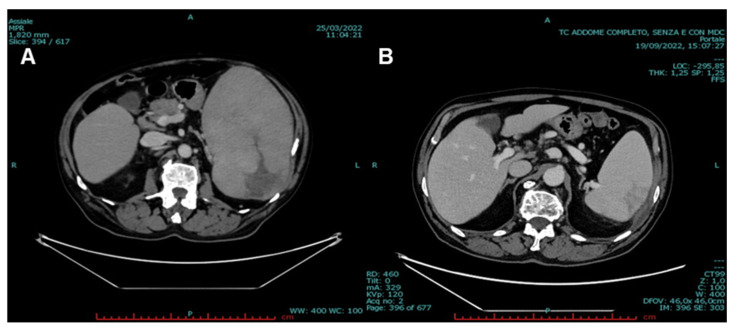
(**A**) CT scan at diagnosis showing spleen enlargement (lateral × transversal diameter, 21 × 12 cm) with hypodense, irregular parenchymal areas. (**B**) CT scan re-evaluation after six months of therapy showing a significant reduction in spleen size (lateral diameter, 12 cm) and in hypodense parenchymal areas.

**Figure 3 jcm-12-00193-f003:**
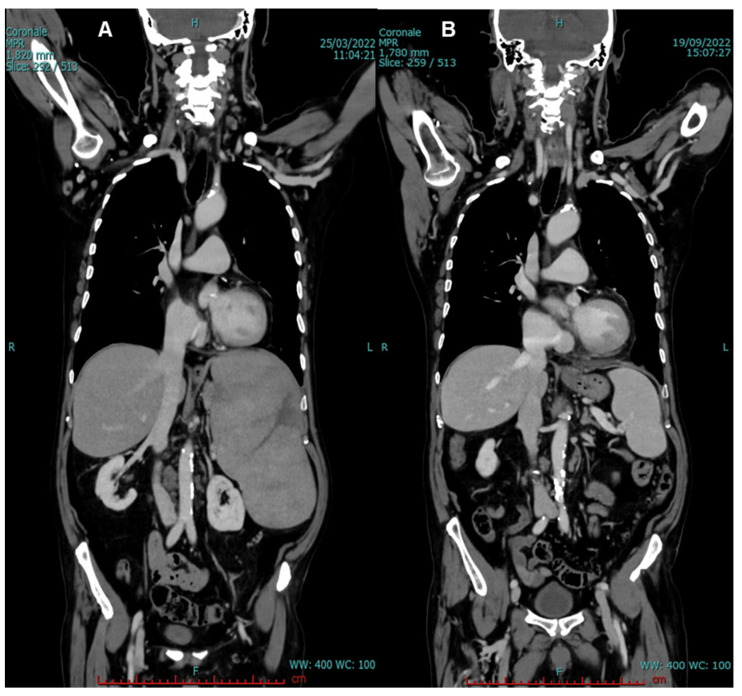
(**A**) Coronal CT scan at baseline, documenting spleen enlargement, and (**B**) significantly reduced after six months of therapy.

## Data Availability

Data are available upon request by the authors.

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
