# Peer review of "A Frail Hairy Cell Leukemia Patient Successfully Treated with Pegylated Interferon-α-2A"

_jcm, 2022, doi:10.3390/jcm12010193_

Round 1

Reviewer 1 Report

Dear authors,

I think it is important to mention ibrutinib in new novel targeted therapies in the introduction.

I would appreciate it if the treatment is more detailed: we know it was started at a dose of 90 microgrammes once a week but was it continued at the same dosage and page? for how long?

Event if it were not appropriate for this patient (chronic atrial fibrillation and mild heart failure), I would discuss the place of ibrutinib as a treatment option for patients who are not eligible for chemotherapy. K. Rogers et al published a phase 2 study of ibrutinib in classic and variant HCL in Blood in 2021 and there is an update of this essay at the ASH (#2891).

Best regards

Author Response

Reviewer 1

Dear authors,

Comment 1. I think it is important to mention ibrutinib in new novel targeted therapies in the introduction.

Response to Comment 1. We thank the Reviewer for this suggestion.

On pages 1-2, lines 44-50, the following text was added “However, several other novel targeted therapies are under investigation for treatment of relapsed/refractory HCL patients, such as recombinant anti-CD22 immunotoxin moxetumomab pasudox, dabrafenib plus trametinib, or Bruton’s tyrosine kinase (BTK) inhibitors. Ibrutinib, an oral BTK inhibitor, administered as single agent has shown prolonged clinical benefits in relapsed HCL patients with an overall response rate of 54%, an estimated 36-month progression-free survival of 73%, and an overall survival OS of 85%, avoiding rituximab while preferring oral drugs [14-16].

The following reference was also added.

  • Rogers, K.A.; Andritsos, L.A.; Wei, L.; McLaughlin, E.M.; Ruppert, A.S.; Anghelina, M.; Blachly, J.S.; Call, T.; Chihara, D.; Dauki, A.; Guo, L.; Ivy, S.P.; James, L.R.; Jones, D.; Kreitman, R.J.; Lozanski, G.; Lucas, D.M.; Ngankeu, A.; Phelps, M.; Ravandi, F.; Schiffer, C.A.; Carson, W.E.; Jones, J.A.; Grever, M.R. Phase 2 study of ibrutinib in classic and variant hairy cell leukemia. Blood. 2021, 137, 3473-3483.

Comment 2. I would appreciate it if the treatment is more detailed: we know it was started at a dose of 90 microgrammes once a week but was it continued at the same dosage and page? for how long?

Response to Comment 2. We apologize for this missing information.

On page 3, lines 116-118, the following text was added “No adverse drug-related reactions with dose reduction or adjustments were documented, and PEG-interferonα-2A was continued at the same dosage of 90 µg once a week.”

On page 4, lines 131-132, the following text was added “At the time of writing, after one year of therapy, the patient is still alive in modified complete response, and he is continuing PEG-interferonα-2A administration

Comment 3. Even if it were not appropriate for this patient (chronic atrial fibrillation and mild heart failure), I would discuss the place of ibrutinib as a treatment option for patients who are not eligible for chemotherapy. K. Rogers et al published a phase 2 study of ibrutinib in classic and variant HCL in Blood in 2021 and there is an update of this essay at the ASH (#2891).

Response to Comment 3. We thank the Reviewer for this suggestion.

On page 3, lines 100-103, the following text was added “Other treatment options were discussed with the patient who expressed the will to avoid intravenous therapies; therefore, monoclonal antibodies were excluded. Moreover, tyrosine kinase inhibitors were not indicated because of cardiovascular comorbidities.”

On page 5, lines 190-194, the following text was added “However, TKIs, including ibrutinib, were not indicated because of cardiovascular comorbidities, especially atrial fibrillation [16]. Therefore, we chose the best possible therapeutic strategy for this patient by balancing clinical efficacy and safety and long-term complications, even though PEG-interferonα-2A could induce a slower and suboptimal disease remission.”

Reviewer 2 Report

General considerations:

-        Extensive editing of English language and style required.

-        Reporting on frontline IFNalpha-2a effectiveness and tolerability as single agent in HCL is relevant; however, the setting of potential use of this approach should be better defined and discussed since other “low intensity”, fast-acting options are available. More insight into IFN-induced immune reconstitution, monitoring and managing toxicities, schedule and duration of therapy should also be given.

Comments:

Line 12: “frail” entails the concept of “unfit”, which might be omitted. Please specify whether you refer to a specific evaluation score when you describe the patient as “frail”.

Line 19: The definition of “salvage therapy” is not appropriate in this setting

Line 66: Can you provide more detailed information on the lymphocyte subset/NK pattern on immunophenotyping at onset et after hematological recovery on IFN therapy? Can you indicate the time required to obtain T lymphocyte subset reconstitution?

Lines 70-73: This description refers to the morphological examination only (and not to the FC immunophenotyping, which you describe afterward.

Line 90: A contraindication to purine analogues is the recent infection in the setting of severe neutropenia, when the profound immunosuppressive side effects of PNA must be avoided, rather than the severe neutropenia itself.

Discussion:

-        Hematological improvement here reported after IFNalpha-2 a single agent turned out to be very slow. Full recovery of neutrophil count and decrease in the spleen size were reported after 6 months of treatment. Such a slow kinetics of response would not be suitable in the event of severe active infection. This therapeutic option should be limited to non life threatening conditions that do not require fast immune recovery. In the event that a faster recovery is required, Rituximab single agent, or a combination approach should probably be considered, leading to a faster response with acceptable toxicity. The benefit of a rapid response to Rituximab probably overcomes Rituximab-related long-term infectious risk. Considerable infectious risk is, on the other hand, expected in the setting of a prolonged severe neutropenia. Please discuss.

-        Please comment on the expected hematological and extra-hematological toxicities of IFN and how you monitored the potential side effects. Indicate whether prolonged severe neutropenia was complicated by infectious events.

-        Please provide justification for IFNalpha-2a dose selection and indicate whether dose adjustments were needed. Given the good tolerability, did you consider increasing the dose to enhance and fasten the hematological response?

-        Please further discuss the outcomes of single agent IFNalpha as frontline therapy in HCL in terms of quality and duration of response, time to treatment failure and rate of discontinuations/dose reductions AEs-related.

-        Please provide more insight into the mechanism of action of IFN, including immunomodulatory effects, and the potential synergistic effect in combination with different agents which might improve IFN efficacy and favor a faster response.

-        Did the patient receive G-CSF to favor ANC recovery? The administration of G-CSF in association with IFNalpha in patients with HCL has been reported for managing or preventing neutropenic complications in the early phases of treatment.

-        Did you consider switching to a different agent or to a drug combination after resolution of severe neutropenia and improvement of performance status, with the aim of improving the quality of response and/or prolong the duration of remission? Discuss the possibility of using PEG-IFNalpha2a as a bridging to potentially more active agents/combinations.

-        Can you specify whether therapy will be given until progression or do you consider a limited-duration therapy to be appropriate? What do you consider to be the goal of IFN single agent therapy that might enable therapy discontinuation?

Author Response

Reviewer 2

General considerations:

Comment 1. Extensive editing of English language and style required.

Response to Comment 1. We have revised our manuscript for English language editing.

Comment 2. Reporting on frontline IFNalpha-2a effectiveness and tolerability as single agent in HCL is relevant; however, the setting of potential use of this approach should be better defined and discussed since other “low intensity”, fast-acting options are available. More insight into IFN-induced immune reconstitution, monitoring and managing toxicities, schedule and duration of therapy should also be given.

Response to Comment 2. We thank the Reviewer for this comment, and we have implemented our discussion by adding mechanisms of action and toxicity management.

On page 5, lines 151-163, the following text was added “Mechanism of action of interferon-α in HCL is still under debate and is likely related to the ability of inducing cell differentiation, growth arrest, and apoptosis [22]. IFNα binds to surface receptors (e.g., IFNAR1) leading to activation of Janus kinases (JAK)/signal transducer and activator of transcription (STAT) pathways and to ex-pression of antiproliferative and proapoptotic genes, including caspases and Fas/CD95 [23]. IFNα can also directly inhibit hematopoietic stem cell growth mainly by blocking JAK/STAT signaling, or indirectly by reducing the production of stimulatory factors, such as IL-11 [24]. These effects can reduce leukemic cell growth, and also cause myelosuppression as drug-related side reaction [23-24]. Moreover, IFNα can promote tumor cell killing and clearance by CD8+ T cells, macrophages, and NK cells, increase the expression of tumor-associated antigens, and enhance antigen presentation by dendritic cells [24-25]. These immunomodulatory activities might be augmented in the presence of TKIs, such as imatinib in chronic myeloid leukemia, favoring immunosuppression and disease progression [25].”

On page 5, lines 172-177, the following text was added “Moreover, HCL patients treated with interferon-α have durable responses, even in follow-up period longer than seven years [28-31]. Reported discontinuation rates due to unmanageable side effects are very low (<20%), even though interferon-α 2a treatment is associated with various side effects, including hematological and extra-hematological toxicities, such as flu‐like syndrome, endocrine and autoimmune disorders, retinopathies, liver, cardiovascular and neuropsychiatric complications [31].”

Comment 3. Line 12: “frail” entails the concept of “unfit”, which might be omitted. Please specify whether you refer to a specific evaluation score when you describe the patient as “frail”.

Response to Comment 3. We agree with the Reviewer, and we have removed unfit from the title and manuscript where not appropriate.

Patient’s frailty was evaluated according to the frailty index, and patient’s score was added in the revised version of our manuscript.

On page 2, lines 70-73, the following text was added “Patient’s frailty was evaluated using the frailty index (FI), defined as FI score = Number of health deficits present / Number of health deficits measured, showing a mild frailty (FI score, 0.332) [19-20].”

The following references were also added.

  • Kojima, G.; Iliffe, S.; Walters, K. Frailty index as a predictor of mortality: a systematic review and meta-analysis. Age Ageing. 2018, 47, 193-200.
  • Peña, F.G.; Theou, O.; Wallace, L.; Brothers, T.D.; Gill, T.M.; Gahbauer, E.A.; Kirkland, S.; Mitnitski, A.; Rockwood, K. Comparison of alternate scoring of variables on the performance of the frailty index. BMC Geriatr. 2014, 14, 25.

Comment 4. Line 19: The definition of “salvage therapy” is not appropriate in this setting

Response to Comment 4. We agree with the Reviewer, and we have removed “salvage” therapy from the entire manuscript.

Comment 5. Line 66: Can you provide more detailed information on the lymphocyte subset/NK pattern on immunophenotyping at onset et after hematological recovery on IFN therapy? Can you indicate the time required to obtain T lymphocyte subset reconstitution?

Response to Comment 5. We thank the Reviewer for this comment, and we have added this missing information of flow cytometry immunophenotyping performed ad diagnosis and post-treatment.

On page 2, lines 83-87, the following text was added “No other alterations were observed, as the remaining 27% of lymphocytes was represented by: 78% of CD3+ T lymphocytes; 13% of CD19+ B cells with no κ/λ restriction (κ/λ, 61%/39%); 24% of CD3-CD56+ Natural Killer (NK) cells; and 15% of CD3+CD56+ NKT cells. CD14+ monocytes were slightly reduced (0.3% of total nucleated cells analyzed).

On pages 3-4, lines 122-130, the following information and discussion were added “No other alterations were observed, and T cell subpopulation frequencies were similar to those reported at baseline (22% vs 27% of total lymphocytes at three months of therapy vs baseline; 75% vs 78% of CD3+ T lymphocytes, respectively; and 13% vs 15% of NKT cells). Conversely, CD19+ B cells were decreased (3% vs 13%, three months of therapy vs baseline), while CD14+ monocytes were increased (7% vs 0.3% of total nucleated cells analyzed). Decreased B cells after treatment might mirror a reduction in certain B cell subpopulations, such as B regulatory cells with immunomodulatory functions, and might be related to physiological immunological reconstitution and restored immune tolerance, as documented in other hematological malignancies, including Hodgkin lymphoma [21].”

The following reference was also added.

  • Giudice, V.; Pezzullo, L.; Ciancia, G.; D'Addona, M.; D'Alto, F.; Gorrese, M.; Cuffa, B.; Selleri, C. Post-therapy B Regulatory Cells Might early Predict Relapse in Hodgkin Lymphoma. Mediterr J Hematol Infect Dis. 2022, 14, e2022042.

Comment 6. Lines 70-73: This description refers to the morphological examination only (and not to the FC immunophenotyping, which you describe afterward.

Response to Comment 6. We apologize for this incorrect sentence, and we have rephrased the sentence as follows “Morphological analysis was performed on peripheral blood showing the presence of 50% of lymphoid cells with intermediate-high nucleus/cytoplasm ratio, loose chromatin, rare nucleoli, basophil cytoplasm, and peripheral hairy protrusions (Figure 1). Flow cytometry immunophenotyping was also performed on peripheral blood evidencing a B-cell clonal population positive for CD20, CD22, CD25, CD11c, CD103, and CD123, and negative for CD5, CD10, CD38 (60% of nucleated cells analyzed).”

Comment 7. Line 90: A contraindication to purine analogues is the recent infection in the setting of severe neutropenia, when the profound immunosuppressive side effects of PNA must be avoided, rather than the severe neutropenia itself.

Response to Comment 7. We thank the Reviewer for this clarification.

On page 3, lines 98-100, the sentence was rephrased as follows “Purine analogues were not indicated, because of recent pneumonia in the setting of severe neutropenia, older age, and comorbidities, including chronic atrial fibrillation, mild heart failure and type 2 diabetes.”

Discussion:

Comment 8. Hematological improvement here reported after IFNalpha-2 a single agent turned out to be very slow. Full recovery of neutrophil count and decrease in the spleen size were reported after 6 months of treatment. Such a slow kinetics of response would not be suitable in the event of severe active infection. This therapeutic option should be limited to non life threatening conditions that do not require fast immune recovery. In the event that a faster recovery is required, Rituximab single agent, or a combination approach should probably be considered, leading to a faster response with acceptable toxicity. The benefit of a rapid response to Rituximab probably overcomes Rituximab-related long-term infectious risk. Considerable infectious risk is, on the other hand, expected in the setting of a prolonged severe neutropenia. Please discuss.

Response to Comment 8. We thank the Reviewer for this point of discussion.

On page 5, lines 178-188, the following text was added “Our patient arrived at our observation in a life-threating condition, due to recent pneumonia in the setting of severe neutropenia, and severe splenomegaly with high risk of rupture. Splenectomy, although not curative, was proposed and refused by the patient. Purine analogues were contraindicated because of high risk of worsening of clinical conditions, because of the recent pneumonia, severe neutropenia, older age, and comorbidities. Our patient also refused intravenous medications, and monoclonal antibodies, including rituximab, could not be used for induction. Moreover, rituximab was not chosen, because of long-term B-cell deficiency leading to several infections, including severe coronavirus-19 disease (COVID-19), even though rituximab in monotherapy or in combination with other drugs induces a faster disease remission and hematological recovery with acceptable toxicity [32-34].”

Comment 9. Please comment on the expected hematological and extra-hematological toxicities of IFN and how you monitored the potential side effects. Indicate whether prolonged severe neutropenia was complicated by infectious events.

Response to Comment 9. We have indicated this information on page 6, lines 205-211 as follows “Expected hematological side effects were close monitored with weekly complete blood counts to exclude transient worsening of drug-related neutropenia. Indeed, following this drug dosage and weekly monitoring, our patient did not experience drug-related adverse events or infectious events. Other extra-hematological toxicities were also monitored with periodic blood tests for endocrine and liver functions, and for autoimmunity [37]. Flu-like effects were significantly reduced using a premedication with acetaminophen before PEG-interferon-α-2A administration.”

Comment 10. Please provide justification for IFNalpha-2a dose selection and indicate whether dose adjustments were needed. Given the good tolerability, did you consider increasing the dose to enhance and fasten the hematological response?

Response to Comment 10. We have considered to increase PEG-interferon-α-2A dosage; however, the patient had a progressive and fairly fast response and we did not want to increase the risk of side effects.

On page 5, lines 194-200, the following text was added “For these reasons, we did not opt for PEG-interferon-α-2A and rituximab combination, differently from a previously reported case [27], thus treatment with PEG-interferon-α-2A in monotherapy was initiated at the lowest weekly dosage (90 µg vs 135 µg), based on previously published studies showing efficacy and safety in other hematological malignancies, such as myeloproliferative neoplasms, using this dosage [35]. Dose increase was not performed because the patient showed hematological response and had no side effects at that PEG-interferon-α-2A dosage.”

Comment 11. Please further discuss the outcomes of single agent IFNalpha as frontline therapy in HCL in terms of quality and duration of response, time to treatment failure and rate of discontinuations/dose reductions AEs-related.

Response to Comment 11. Please refer to Response to Comment 2.

Comment 12. Please provide more insight into the mechanism of action of IFN, including immunomodulatory effects, and the potential synergistic effect in combination with different agents which might improve IFN efficacy and favor a faster response.

Response to Comment 12. Please refer to Response to Comment 2.

Comment 13. Did the patient receive G-CSF to favor ANC recovery? The administration of G-CSF in association with IFNalpha in patients with HCL has been reported for managing or preventing neutropenic complications in the early phases of treatment.

Response to Comment 13. Although we initially considered the use of G-CSF, its use was ruled out because of splenomegaly and imminent risk of splenic rupture.

On page 3, lines 107-108, the following text was added “No growth factors were administered for controlling the neutropenia because of severe splenomegaly and high risk of splenic rupture.”

Comment 14. Did you consider switching to a different agent or to a drug combination after resolution of severe neutropenia and improvement of performance status, with the aim of improving the quality of response and/or prolong the duration of remission? Discuss the possibility of using PEG-IFNalpha2a as a bridging to potentially more active agents/combinations.

Response to Comment 14. We thank the Reviewer for this point of discussion. We have considered to switch to other therapies; however, the patient achieved a modified complete remission without significant side effects, therefore, we decided to continue PEG-IFNalpha2a, especially because the patient refused intravenous therapies. If hematologic response would have been incomplete or patient’s conditions not improved, the use of interferon as a bridge to alternative therapies would have been adopted.

On page 6, lines 216-221, the following text was added “However, limited evidence of PEG-interferon-α-2A treatment duration in HCL is available, and therapeutic strategy might be modulated based on clinical response, by adding an additional oral drug or an intravenous monoclonal antibody, such as rituximab, if patient shows refractoriness or disease relapse, or by reducing dose or administration frequency or by interferon discontinuation if a very profound and stable CR is achieved.”

Comment 15. Can you specify whether therapy will be given until progression or do you consider a limited-duration therapy to be appropriate? What do you consider to be the goal of IFN single agent therapy that might enable therapy discontinuation?

Response to Comment 15. Please refer to Response to Comment 14.

Round 2

Author Response

Title: “Literature review” is probably not consistent with the content of the paper, as data regarding frontline PEG IFN alpha2a in HCL are extremely limited.

Response. The title was changed as follows “A frail hairy cell leukemia patient successfully treated with pegylated interferon-? 2A”

Abstract:

Line 13: delete “according to frailty index”

Line 15 “with PEG IFN alpha2a in monotherapy” rather than “only with”

Response. “according to frailty index” was removed and, on line 15 the text was changed as “with PEG IFN alpha2a in monotherapy”.

Introduction:

Line 43: delete “moreover”

Response. Deleted.

Line 50: delete “avoiding Rituximab while preferring oral drug”

Response. Deleted.

Line 52: what do you mean by high-dose chemotherapy? Do you mean PNA therapy? That could be misinterpreted , please specify.

Response. We apologize for this misleading sentence that was rephrased as follows “or not eligible for purine analogues”.

Line 54: delete “unfit”

Response. Deleted.

Case presentation:

Line 59: 82-year-old male

Response. Corrected.

Lines 84-85; 122-125: absolute values and T CD4/T CD8+ ratio should also be specified

Response. CD4/CD8 ratio was not reported at diagnosis, while values at follow-up were added on line 124 (CD4+/CD8+ cells, 55%/19%). Absolute counts of CD4+/CD8+ by flow cytometry were not performed.

Lines 99 and 103: “contraindicated” rather than “not indicated”

Response. Corrected.

Line 107: “Growth factors were not administered to favor ANC recovery” rather than “for controlling the neutropenia”

Response. Corrected.

Lines 118-119: “no drug-related reactions requiring dose reduction or discontinuation adjustement”.

Response. Corrected.

Delete “and PEG IFNalpha-2a was continued at the same dosage…”

Response. Deleted.

Lines 122-125: delete and replace with “No alterations were observed in the lymphocyte subsets

except for a decrease in CD19+ B cells and an increase in CD14+ monocytes” (absolute values?)

Any modification in absolute values and T CD4/T CD8+ ratio?

Response. The entire sentence was rephrased as follows “No alterations were observed in the lymphocyte subsets except for a decrease in CD19+ B cells and an increase in CD14+ monocytes (CD3+ T lymphocytes, 75% with CD4+/CD8+ cells, 55%/19%; NKT cells, 13%; CD19+ B cells, 3%; and CD14+ monocytes 7%).

Line 129: “restored immune tolerance” is not consistent with the reduction of B regulatory cells

that you mentioned. Please clarify or delete.

Response. Deleted.

Line 137: Is there general agreement on the definition of “modified” complete response? If not, please specify that the definition of CR according to ESMO guidelines requires the absence of hairy cells on peripheral blood and bone marrow aspiration or biopsy specimen

Response. The term “modified” was removed.

Line 138; “ in the absence of bone marrow aspiration or biopsy reassessment” rather than

“without repetition…”

Response. Corrected.

Discussion:

Line 162-164: delete, the content is in contrast to what expected in the presence of the above mentioned immunomodulatory effects

Response. Deleted.

Please provide more insight into the potential synergistic effect in combination with different agents which might improve IFN efficacy and favor a faster response (i.e.: the administration of IFNα-2a before and during rituximab treatment could be effective in increasing CD20 antigen surface expression on neoplastic B cells; IFNα-2a-induced stimulation of T cell cytotoxicity and NK cell activity might favor antibody-dependent cell mediated cytotoxicity).

Response. On page 5, lines 170-177, the following text was added “Interferon-α in monotherapy could be used as a priming therapeutic phase followed by combination with rituximab with good tolerability even in immunosuppressed HCL patients [32]. During the priming phase, interferon-α-2a could increase CD20 antigen expression on hairy cells, and simultaneously stimulate CD8+ T cell and NK cytotoxic activity. Therefore, when rituximab is combined with interferon-α-2a during the second phase, tumor cell killing is enhanced likely through antibody-dependent cell mediated cytotoxicity, as demonstrated by clinical efficacy of interferon-α-2a and rituximab in low-grade lymphomas [32-34].”

The following references were also added.

  1. Furlan, A.; Rossi, M.C.; Gherlinzoni, F.; Scotton, P. Prompt Hematological Recovery in Response to a Combination of Pegylated Interferon α-2a and Rituximab in a Profoundly Immuno-Suppressed Hairy Cell Leukemia Patient with a Mycobacterial Infection at Onset: Benefits and Drawbacks of Rapid Immune Reconstitution. Hematol Rep. 2022, 14, 135-142.
  2. Gidlund, M.; Orn, A.; Wigzell, H.; Senik, A.; Gresser, I. Enhanced NK cell activity in mice injected with interferon and interferon inducers. Nature. 1978, 273, 759-61.
  3. Sacchi, S.; Federico, M.; Vitolo, U.; Boccomini, C.; Vallisa, D.; Baldini, L.; Petrini, M.; Rupoli, S.; Di Raimondo, F.; Merli, F.; Liso, V.; Tabilio, A.; Saglio, G.; Vinci, G.; Brugiatelli, M.; Dastoli, G.; GISL. Clinical activity and safety of combination immunotherapy with IFN-alpha 2a and Rituximab in patients with relapsed low grade non-Hodgkin's lymphoma. Haematologica. 2001, 86, 951-8.

Line 166: “IFN efficacy was subsequently confirmed…”

Response. Corrected.

Lines 171-173: please specify if you refer to the frontline setting

Response. On page 5, lines 166-168, the sentence was rephrased as follows “Interferon-α also shows high efficacy as first line therapy in patients aged >65 years with comorbidities, or in those resistant to purine analogues, with a 5-year PFS of 68-96% [30].”

Lines 175: IFN-alpha 2a is potentially associated with hematological and extra-hematological… delete “various side effects, including”

Response. Corrected and deleted.

Lines 192-195 “Even though IFN-alpha 2a monotherapy potentially induces a slower and suboptimal disease response, we chose…”

Response. Corrected.

Lines 198-199: “showing efficacy and safety of this dosage…”. Delete “using this dosage”

Response. Deleted.

Line 201: “and good tolerability” rather than “and had no side effects…”

Response. Corrected.

Line 202: “more favorable pharmacokinetic profile”

Response. Corrected.

Lines 205-208 “hematological side effects were closely monitored”; delete “to exclude transient worsening of drug-related neutropenia”, as neutropenia was already severe at onset.

Response. Corrected.

Delete “Indeed, following this drug dosage and weekly monitoring, our patient did not experience drug-related adverse events or infectious events”.

Response. Deleted.

Add a comment regarding the absence of infectious complications in the case description section (i.e. after line 120)

Response. On page 3, line 119, the following text was added “No infectious complications were also observed.”

Line 212: “in inducing a hematological improvement…”

Response. Corrected.

Delete “moreover”

Response. Deleted.

Line 216: “limited data regarding optimal treatment duration…”: “the therapeutic strategy might be modulated based on clinical response and tolerance…”

“considering the association of an additional agent … in the event of refractoriness or relapse; a reduction in dosage or frequency of administration or drug discontinuation if a profound and stable remission is achieved.

Response. Corrected.